# Barriers to healthcare access among women in sub-Saharan Africa: A pooled analysis of multi-country DHS data (2019–2023)

Gebreeyesus Abera Zeleke [1]*, Enyew Getaneh Mekonen[1], Bewuketu Terefe[2], Mulugeta Wassie[3], Agazhe Aemro[3], Alebachew Ferede Zegeye[3], Mohammed Seid Ali[4], Berhan Tekeba[4], Tadesse Tarik Tamir[4], Belayneh Shetie Workneh[5]

1 Department of Surgical Nursing, School of Nursing, College of Medicine and Health Sciences, University of Gondar, Gondar, Ethiopia, 2 Department of Community Health Nursing, School of Nursing, College of Medicine and Health Sciences, University of Gondar, Gondar, Ethiopia, 3 Department of Medical Nursing, School of Nursing, College of Medicine and Health Sciences, University of Gondar, Gondar, Ethiopia, 4 Department of Pediatrics and Child Health Nursing, School of Nursing, College of Medicine and Health Sciences, University of Gondar, Gondar, Ethiopia, 5 Department of Emergency and Critical Care Nursing, School of Nursing, College of Medicine and Health Sciences, University of Gondar, Gondar, Ethiopia

* gebreeyesusabe143@gmail.com

## Abstracts

### Background

Barriers to accessing healthcare have significantly contributed to the high rates of maternal and child mortality and morbidity in developing regions, particularly in Sub-Saharan Africa. Access to healthcare is influenced by multiple factors, including financial constraints, geographic location, the availability of services, and the quality of care provided. This study aimed to assess the magnitude of these barriers and to identify the factors influencing healthcare access among women of reproductive age in Sub-Saharan Africa.

### Methods

This study utilized secondary data from the most recent Demographic and Health Surveys conducted between 2019 and 2023 across Sub-Saharan Africa. A weighted sample of 134,470 women of reproductive age was included in the analysis. Data were analyzed using Stata version 14, employing a multilevel mixed-effects logistic regression model. Factors associated with barriers to healthcare access were considered statistically significant at P-values less than 0.05. Adjusted odds ratios (AORs) with 95% confidence intervals (CIs) were used to interpret the results.

### Results

In this study, the prevalence of barriers to healthcare access among women found to be 55.84% (95% CI:55.57, 56.10). Husband educational status (AOR=1.23, 95%

**Data availability statement:** All relevant data are within the manuscript and its Supporting information files.

**Funding:** The author(s) received no specific funding for this work.

**Competing interests:** The authors have declared that no competing interests exist.

CI: 1.15, 1.30), media exposure (AOR=1.17, 95% CI: 1.11, 1.24), antenatal care visit (AOR=1.31,95% CI: 1.10, 1.55), delivery place (AOR=1.33, 95% CI:1.24, 1.43), health insurance (AOR=1.18, 95% CI: 1.11, 1.26), wealth index(AOR=2.29, 95% CI 2.14, 2.45), (AOR=1.57, 95CI: 1.46, 1.68) and visiting health facility within the past 12 months (AOR=0.91,95% CI:0.87, 0.97) were individual level factors associated with outcome variable. In addition, place of residence (AOR=1.30, 95% CI:1.22, 1.39), community literacy level (AOR=1.13, 95% CI:1.03, 1.24), community poverty level (AOR=1.12, 95% CI:1.02, 1.22), and Region were community level factors associated with barriers to healthcare access.

## Conclusion

In this study, over half of women in Sub-Saharan Africa face significant barriers to healthcare access. Educational status, media exposure, place of residence, antenatal care attendance, place of delivery, health insurance coverage, wealth index, recent visits to health facilities, community literacy levels, community wealth status, and regional differences were determinant factors. These findings highlight the need for Sub-Saharan African governments to prioritize these determinants when formulating policies and strategies aimed at reducing preventable barriers to healthcare access in the region.

## Introduction

Access to healthcare is broadly defined as the ability to prevent illness, disability, and premature death, to promote and maintain health, and to attain health equity, particularly for women of reproductive age, by offering inclusive, high-quality care and the best possible availability, affordability, and accessibility of medical services [1–3]. Reproductive-age women's health is a critical global public health concern since it influences population health outcomes and fosters socioeconomic growth [4]. The advancement of socioeconomic development is significantly influenced by the health of women of reproductive age, especially in Africa [5]. The Sustainable Development Goals (SDGs) place a strong emphasis on women's health, with Targets 3.7 and 3.8 of SDG 3 specifically focusing on achieving universal health coverage by 2030 and ensuring access to sexual and reproductive health services, including education and information on family planning [6,7]. The Sustainable Development Goals (SDGs) explicitly target a substantial reduction in maternal mortality, with Goal 3.1 aiming to lower the global maternal mortality ratio to fewer than 70 per 100,000 live births by 2030. While these global efforts have contributed to some improvements in women's health outcomes, they have yet to fully address the root causes of maternal mortality. Achieving sustained and comprehensive progress remains a critical challenge. In 2016, Sub-Saharan Africa accounted for a disproportionately high share of the 303,000 maternal deaths reported globally, the majority of which were due to preventable causes. Moreover, non-communicable diseases continue

to contribute significantly to female morbidity and mortality across nearly all countries, with a particularly severe impact in low- and middle-income settings, where many women are left with long-term disabilities [4–6]. Access to healthcare involves multiple dimensions, including financial affordability, geographic accessibility, availability of medical services, and the quality of care provided in health facilities [8]. Access to healthcare is a key determinant of whether individuals utilize health services. Globally, approximately 400 million people lack essential healthcare, and around eight million die each year from preventable diseases—resulting in an estimated economic loss of $6 trillion in low- and middle-income countries [6,9,10]. In 2015, world leaders pledged to achieve universal health coverage as part of the Sustainable Development Goals (SDGs), aiming to ensure access to affordable, high-quality medicines and financial protection for all [11,12]. In 2017, the World Bank and WHO reported that half of the global population lacks access to essential health services. As a result, many countries in Sub-Saharan Africa have made universal healthcare a key part of their national health strategies. However, there has been slow progress in turning this commitment into tangible results, particularly in securing national funding for equitable, high-quality healthcare, financial security, and overall health [6,9,13,14]. Reproductive age women and child mortality from low family planning uptake, unsafe abortion practices, unintended pregnancies, and an increase in home births are all implications of women in reproductive age having trouble accessing health care [14]. Previous research has offered some evidence regarding the sociodemographic and economic factors that can be barriers to accessing healthcare, such as marital status, place of residence, maternal age, wealth index, educational attainment, and unemployment status. Nevertheless, women of reproductive age continue to face significant disparities in healthcare access and Post-COVID-19 evidence on healthcare access remains limited, particularly in low-resource settings; this study addresses gap by leveraging recent DHS data to examine regional disparities, intersectional demographics, and implications for health system recovery and healthcare access [10,14–16]. This study makes three unique contributions to the body of existing literature. In order to provide a current and policy-relevant analysis of the barrier to healthcare access, it first makes use of the most recent 2019–2023Demographic and Health Survey (DHS) data. Second, our analysis takes a harmonized multi-country approach across ten sub-Saharan African nations, improving cross-national comparability and generalizability in contrast to many previous studies that are limited to single-country analyses or rely on older datasets. In order to gain a better understanding of the structural and contextual factors that influence healthcare access, the study employs a comprehensive multilevel analysis of determinants at the individual and community levels. These contributions put our research in a position to guide focused, empirically supported programs provide fair access to healthcare for reproductive age women in sub-Saharan. Therefore, this study aimed to ascertain the magnitude and potential contributing factors associated with barriers to healthcare access among reproductive age women in sub-Saharan Africa.

## Methods

### Study design, study area, and period

Data from ten Sub-Saharan African countries, collected between 2019 and 2023, were used to perform a multilevel mixed-effects analysis, based on the most recent Demographic and Health Surveys (DHS). The DHS collects community-based, cross-sectional data approximately every five years to generate updated health and health-related indicators.

### Patient and public involvement statement

This study was used, Secondary data from the Demographic and Health Surveys data.

### Data source, study population and sampling technique

A secondary data analysis was conducted using Demographic and Health Survey (DHS) data collected between 2019 and 2023 from selected sub-Saharan African countries. The study specifically included DHS datasets from ten countries: Burkina Faso, Gabon, Ghana, Kenya, Liberia, Rwanda, Sierra Leone, Tanzania, The Gambia, and Mozambique, each

                                                

of which had recent and publicly available survey data within the specified timeframe. The datasets were appended to assess the extent of, and factors underlying, barriers to healthcare access among women of reproductive age in sub-Saharan Africa. Each country's DHS comprises multiple datasets. The DHS employs a stratified, two-stage cluster sampling design: in the first stage, enumeration areas are selected, followed by systematic sampling of households within each selected area in the second stage.

The variable barrier to healthcare access was generated by computation of (v467b, v467c, v467d, v467f) from maternal recode (IR) data set was recoded. A binary logistic regression model was used to identify determinants factor associated with barriers healthcare access. Determinants barriers to healthcare access were reported in terms of an AOR with a significance level of (95%) confidence interval. In the univariate analysis, at 95% confidence intervals with a p-value of < 0.25 was considered for the candidate multivariable analysis of data (S1 Table 5). All variables with p values of <0.05 were considered statistically significant. Since DHS data are nationally representative, stratified multistage cluster sampling is required; therefore, sampling weights must be applied to produce unbiased estimates. To correctly weigh DHS data, we used the sampling weights provided in the dataset, specifically the individual weight (v005) for women's individual recode files. This weight must be divided by 1,000,000 (v005/1000000). using "gen wgt=v005/1000000 Stata commands. A total weighted sample of 134,470 reproductive age women were included in the study (Table 1).

## Study variables

**Outcome variables:** The outcome variable of this study was barriers to healthcare access. Based on DHS guidelines, outcome variable was constructed as a composite measure based on four indicators: (1) distance to a health facility, (2) difficulty in obtaining money for treatment, (3) difficulty in obtaining permission to visit a health facility, and (4) reluctance to go to a health facility alone. Women who respond experiencing at least one of these four challenges were classified as facing significant barriers to healthcare access and were coded as 1 and leveled as "big problem". Conversely, women who respond no difficulty in any of the four indicators were categorized as having no significant barriers to healthcare access and were coded as 0 and leveled as "no big problem" [17]. Handling missing value on outcome variables were retained in the denominator accordance with DHS analytical guidelines.

**Individual level Variables:** Hence DHS data are hierarchical, independent factors come from two sources: that was at individual and community levels factors were considered for this analysis. The individual-level independent variables were women's age (15–19, 20–35, 36–49), Women educational status (no education, Primary education, Secondary & Higher education), women occupation (working, not working), Religion (Muslim, Christian, Others), Wealth index (Poor, Middle,

**Table 1. The weighted Sample size for magnitude and determinants of barriers to accessing health care among reproductive age women in sub-Saharan Africa.**

| Country | Year of survey | weighted sample (n) | weighted sample (%) |
|---|---|---|---|
| Burkina Faso | 2021 | 17659 | 13.13 |
| Gabon | 2019-2021 | 6507.15 | 4.83 |
| Ghana | 2022 | 15014 | 11.16 |
| Kenya | 2022 | 16716.36 | 12.43 |
| Liberia | 2019-2020 | 8065 | 5.99 |
| Rwanda | 2019-2020 | 14634 | 10.88 |
| Sierra Leone | 2019 | 15574 | 11.58 |
| Tanzania | 2022 | 15254 | 11.34 |
| Gambia | 2019-2020 | 11865 | 8.08 |
| Mozambique | 2023 | 13181.93 | 9.80 |
| TOTAL | | 134470 | 100% |

Rich) media exposure (yes, No), internet utilization (yes, No), Residency (urban, rural), Husband educational status (No education, Primary education, Secondary& Higher education), Husband occupational status (No working, Working), using Contraception (No, Yes) Visiting health facility in past 12 months (No, Yes), place of delivery (Home, Institutional delivery), ANC visit (No antenatal visit, 1–4 visit, >4 visit).

**The community-level Variables:** Poverty level (Low, High), Literacy level (Low, High), Media exposure, (Low, High), Residency (Urban, Rural), Region (East, West), Community ANC visit.

## Data processing and statistical analysis

Stata version 14 statistical software was used to clean, record, and analyze secondary DHS data sets. The variables in the DHS data are grouped into clusters, and the variables within a cluster are more similar to each other than those in other clusters. The presumptions of equal variance across clusters and independent data were violated in order to run a typical logistic regression model. This suggests that between-cluster factors must be considered using a complex model.

Given this, multilevel mixed-effects logistic regression model was used to determine the factors that associated with barrier to accessing the health care. Multilevel mixed effect logistic regression follows four models. The null model (only outcome variable), mode I (individual level variable such as age, Religious status, educational status, Husband educational, marital occupation, Husband occupation, marital status, Contraceptive Utilization, place of delivery, ANC visit, media exposure, Intermate utilization, visiting health facility within 12 months, health insurance, wealth index) model II community level variables, Region in sub-Saharan Africa, community poverty level, community media exposure, Community distance health facility, Community literacy level, Community ANC visit), and model III (both individual and community level variables such as age, Religious status, educational status, Husband educational, marital occupation, Husband Occupation, marital status, Contraceptive Utilization, place pf delivery, ANC visit, media exposure, Intermate utilization, visiting health facility within 12 months, health insurance, wealth index), model II (community level variables, Region in sub-Saharan Africa, community poverty level, community media exposure, Community distance health facility, Community literacy level, Community ANC visit). The null model was used to check the variability of outcome variable across the cluster. The relationship of individual-level variables with outcome variable (Model I) and the association of community-level variables with outcome variable (Model II) were determined. In the final model, the association of both individual and community-level variables was fitted with the outcome variable.

## Random effects

The intra-class correlation coefficient (ICC), proportionate change in variance (PCV), and median odds ratio (MOR) were used to quantify random effects or measures of variation of the outcome variables.

To assess the variation between clusters, the proportionate change in variance (PCV) and intra-cluster correlation coefficient (ICC) were calculated as By treating the cluster as a random variable, the ICC shows that the variation in the health care access barriers between clusters may be computed as follow $ICC = \frac{\sigma^2 uo}{\sigma^2 uo + \pi^2/3} \times 100\%$. The MOR, which is determined by using clusters as a random variable and calculating the median value of the odds ratio between the highest-risk and lowest-risk areas for barriers to healthcare access was calculated as: $MOR = \exp(0.6745 \sqrt{(2 * \sigma^2 uo)})$ Furthermore, the PCV calculated as: variable that explains the variation in the degree of barriers to healthcare access $PCV = \frac{\sigma^2 uo - \sigma^2}{\sigma^2 uo} \times 100\%;$. where VC is the cluster level variance and Vnull is the variance of the null model.

The associations between independent factors at the individual and community levels and barriers to accessing health care were estimated using the fixed effects. It was assessed and presented using adjusted odds ratio (AOR) and 95% confidence intervals with a p-value of < 0.05. Because of the nested nature of the model, deviation = −2(log likelihood ratio) was used to compare models, and the model with the lowest deviance was selected as the best-fitted model. By evaluating the variance inflation factors, the model assumption for multi-collinearity was confirmed. In addition, Model comparison was performed using Akaike's Information Criterion (AIC). The model includes both random intercepts and

random slopes (Model III) was evaluated alongside alternative models, with AIC values calculated to assess relative model fit and parsimony (S2 Table 6).

### Ethical approval and consent to participate

This study was based on analysis of existing survey datasets in the public domain (SSA) that are freely available online with all the identifier information anonymized, no ethical approval was required.

## Results

In this study a total of 134,470 reproductive age women were included from 10 Sub-Saharan African countries. The Mean age of participants was 28.99 years with SD. 9.68. More than two thirds (69.8%) of women did not use contraception, and over 72.22% reproductive age women did not use health insurance, and over half (54.22%) of women were living in rural areas in sub-Saharan Africa (Table 2).

In this study, the magnitude of barriers to accessing health care among reproductive age women was found to be 55.84 (95% CI: (55.57, 56.10) owing to one or more of the four factors, the most commonly cited being the difficulty in getting money and the distance from medical facilities (Fig 1).

### Random effect (Measures of variation) and model fitness

To determine if the data supported the choice to evaluate randomness at the community level, a null model was employed. With the variance of 0.3784967 and a p-value of < 0.001, the null model's results showed that there were notable variations in the barriers to accessing healthcare among clusters. A significant amount of the variance in the outcome between clusters was jointly explained by both individual- and community-level variables, as evidenced by the intra-cluster correlation (ICC), which dropped from 10.31% in the null model to 3.96% in the fully adjusted model (model III). According to the proportionate change in variance (PCV), Model III explains the greatest percentage (64.09%) of the group-level variation in comparison to the null model, group comparison showed that, Model I (50.19%) and Model II (24.56%). This suggests that the model fits better as more predictors are included. Model comparison using Akaike's Information Criterion (AIC) indicated that Model III achieved the best fit, as evidenced by the lowest AIC (239,608.6), markedly lower than both the null model (AIC = 1,128,437) and Model II (AIC = 1,127,704). Model I also showed good fit (AIC = 239,764.8), though slightly less optimal than Model III and highest log-likelihood value (−21,433.284) among all competing models. Multicollinearity using variance inflation factors (VIF) indicated that all VIF values ranged from 1.03–1.98, which is within accepted range (Table 3).

### Determinants of barriers to healthcare access among reproductive age women in Sub-Saharan Africa

**Multivariable multilevel logistic regression analysis of individual and community level factors associated with Barriers to health care access among women in Sub-Saharan Africa.** In the final fitted model of multivariable multilevel logistic regression, Husband educational status, media exposure, place of residency, Antenatal care visit, place of delivery, health insurance utilization, wealth index status, visiting health facility within 12 months, and country category (western, eastern Africa) were significantly associated with barriers of accessing health care among reproductive age women in sub-Saharan Africa. The odds of barriers of accessing health care were 1.30 times higher among reproductive age women living in rural residency compared to women living in an urban area (AOR = 1.30, 95% CI: 1.22, 1.39). and the odds of barriers healthcare access were 1.23 times more likely to among reproductive age women whose husband had no education compared to women whose husband had secondary and higher education (AOR = 1.23, 95% CI: 1.15, 1.30). In addition, odds of barriers healthcare access were 1.33 times more likely to occur in women who delivered at home

**Table 2. Sociodemographic and economic characteristics of reproductive age women who had barriers to healthcare access in sub-Saharan Africa.**

| Individual and Community level variables | Frequency(n) | Percentage (%) |
|---|---|---|
| **Individual level variables** | | |
| **Maternal age** | | |
| 15-19 | 27890.4 | 20.7 |
| 20-35 | 69441.4 | 51.64 |
| 36-49 | 37138.6 | 27.64 |
| **Religious Status** | | |
| Muslim | 48834.95 | 41.96 |
| Christian | 46856.90 | 39.30 |
| Other | 23524.59 | 19.73 |
| **Educational Status** | | |
| No formal education | 35051.70 | 26.06 |
| Primary education | 39651.17 | 29.48 |
| Secondary and higher education | 59767.57 | 44.44 |
| **Husband's Education** | | |
| No formal education | 29414.44 | 36.72 |
| Primary education | 21245.78 | 26.52 |
| Secondary and Higher education | 29428.78 | 36.74 |
| **Maternal Occupational** | | |
| Not working | 50732.90 | 37.81 |
| Working | 83417.01 | 62.19 |
| **Husband Occupation** | | |
| Not working | 10390.43 | 13.00 |
| Working | 69698.11 | 87.00 |
| **Marital Status** | | |
| Single | 42213.40 | 31.4 |
| Married | 58262.00 | 43.3 |
| Other | 33995.4 | 25.3 |
| **Contraceptive Utilization** | | |
| No | 93855.00 | 69.8 |
| Yes | 40615.5 | 30.2 |
| **Place of Delivery** | | |
| Home | 8279.46 | 15.5 |
| Institution | 44979.28 | 84.5 |
| **ANC visit** | | |
| No visiting | 2012.84 | 3.77 |
| 1–4 visiting | 24459.34 | 45.92 |
| >4 visiting | 26786.55 | 50.29 |
| **Media exposure** | | |
| No | 34628.45 | 25.75 |
| Yes | 99831.57 | 74.24 |
| **Internet utilization** | | |
| No | 93020.34 | 69.17 |
| Yes | 41450.10 | 30.82 |

*(Continued)*

**Table 2.** (Continued)

| Individual and Community level variables | Frequency(n) | Percentage (%) |
|---|---|---|
| **Visited Health Facility Last 12 Months** | | |
| No | 57137.73 | 42.49 |
| Yes | 77332.71 | 57.50 |
| **Health Insurance** | | |
| No | 85044.69 | 72.22 |
| Yes | 32709.39 | 27.77 |
| **Wealth Index** | | |
| Poor | 47227.18 | 35.12 |
| Middle | 25771.45 | 19.16 |
| Rich | 61471.81 | 45.74 |
| **Community level variables** | | |
| **Region in sub-Saharan Africa** | | |
| East Africa | 59786.29 | 44.46 |
| West Africa | 74684.16 | 55.54 |
| **Residency** | | |
| Urban | 61558.03 | 45.78 |
| Rural | 72912.41 | 54.22 |
| **Community poverty level** | | |
| Low | 70776.71 | 52.63 |
| High | 63693.73 | 47.47 |
| **Community media exposure** | | |
| Low | 61642.30 | 45.84 |
| High | 72828.14 | 54.15 |
| **Community distance health facility** | | |
| Low | 65962 | 49.00 |
| High | 68508 | 51.00 |
| **Community literacy level** | | |
| Low | 57615.01 | 42.84 |
| High | 76855.43 | 57.15 |
| **Community ANC visit** | | |
| Low | 77912.32 | 57.96 |
| High | 56491.92 | 42.03 |

compared to women who had institutional delivery (AOR = 1.33, 95% CI: 1.24, 1.43), and the odds of barriers healthcare access were 1.31 times higher among women who had no history of antenatal visit compared to women who had more than four ANC visit (AOR = 1.31, 95% CI: 1.10, 1.55); The odds of barriers to healthcare access among reproductive age women who had no media exposure were 1.17 times more likely to compared with a woman who had media exposure (AOR = 1.17, 95% CI: 1.11, 1.24) and the barriers to healthcare access among reproductive age women who were not visiting a health facility within 12 months were 9% times less likely compared to women who had visiting health care facility within 12 months of study period (AOR = 0.91, 95% CI: 0.87, 0.97). The odds of perceived barrier of accessing health care among reproductive age women who had no health insurance utilization were 1.18 times more likely to compared with a woman who had used health insurance (AOR = 1.18, 95% CI: 1.11, 1.26). additionally, the odds of perceived barriers healthcare access among

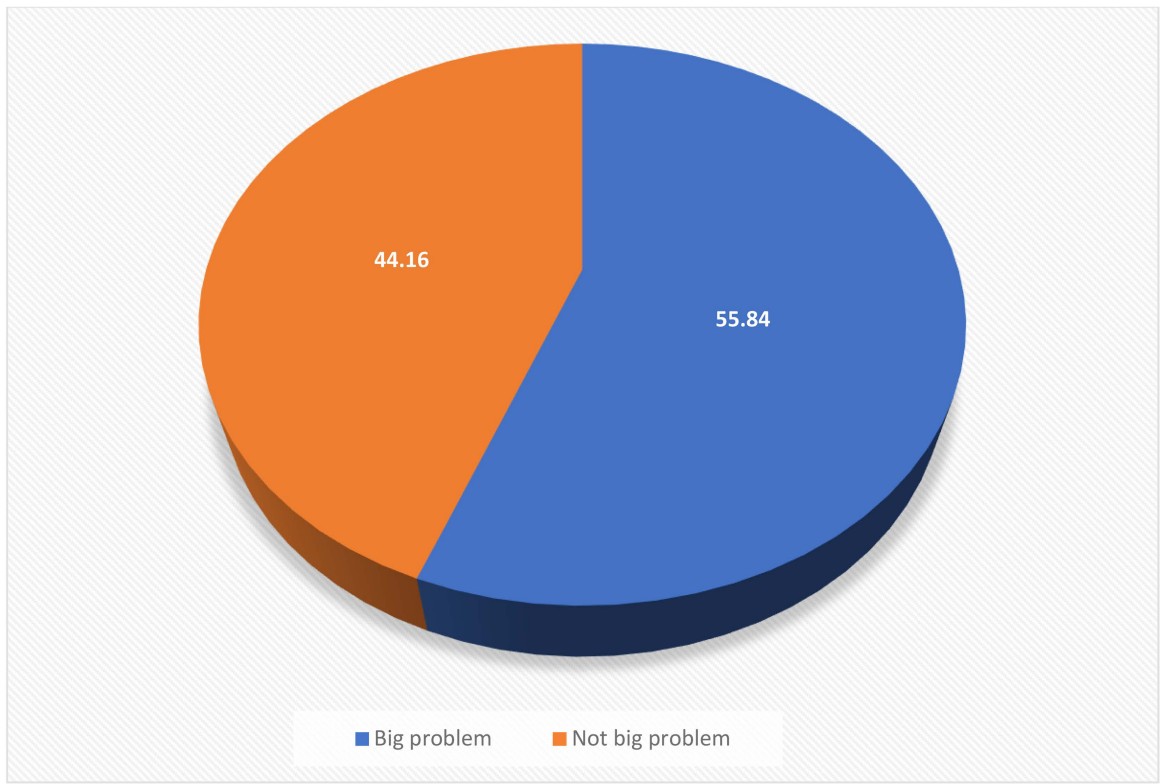

**Fig 1. Barriers to accessing health care and determinant factor among reproductive age women in sub-Saharan Africa.**

**Table 3. Model comparison and Fitness random effect analysis for barriers of accessing health care of among reproductive age women in sub-Saharan Africa.**

| Parameter | Null model | model I | model II | model III |
|---|---|---|---|---|
| Variance | 0.3784967 | 0.1885269 | 0.28553 | 0.1358817 |
| ICC | 10.31% | 5.41% | 7.98% | 3.96% |
| MOR | 1.79 | 1.51 | 1.66 | 1.41 |
| PCV | Reference | 50.19% | 24.56% | 64.09% |
| Model Fitness | | | | |
| LLR | −90175.365 | −21798.27 | −86841.704 | −21433.284 |
| Deviance | 1803596.73 | 43596.54 | 173683.408 | 42866.568 |
| AIC | 1128437 | 239764.8 | 1127704 | 239608.6 |

**ICC:** intra-cluster correlation, **LLR:** loglikelihood ratio, **MOR:** median odds ratio, **PCV:** proportional change in variance. **AIC:** Akaike information criteria.

reproductive age women who were classified under poor and middle wealth status were 2.29 and 1.57 times more likely of barriers to healthcare access compared to reproductive age women who were classified under rich wealth status category (AOR = 2.29 95% CI:2.14, 2.45) and (AOR = 1.57, 95% CI: 1.46, 1.68). The odds of a perceived barrier of accessing health care service among reproductive age women who live in West Africa region were 2.38 times more likely barriers to healthcare access compared to those living in East Africa region (AOR = 2.38, 95% CI: 2.21, 2.55) (Table 4).

**Table 4. Multivariable multilevel logistic regression of individual and community level factors associated with barriers to healthcare access among reproductive age women in Sub-Saharan Africa.**

| Individual and community level factors | Model I AOR [95% CI] | Model II AOR [95% CI] | Model III AOR [95% CI] |
|---|---|---|---|
| **Individual level factors** | | | |
| **Maternal age** | | | |
| 15-19 | 0.96 [0.86, 1.07] | | 0.96 [0.86,1.08] |
| 20-35 | 1.06 [1.01,1.13] | | 1.04 [0.99, 1.11] |
| 36-45 | 1 | | 1 |
| **Residency** | | | |
| Urban | | 1 | 1 |
| Rural | | 2.29 [2.24, 2.35] | 1.30 [1.22, 1.39] * |
| **Maternal education** | | | |
| No education | 1.10 [1.02, 1.18] | | 1.04 [0.97, 1.11] |
| Primary education | 0.93 [0.87, 0.99] | | 1.06 [0.99, 1.14] |
| Secondary and above | 1 | | 1 |
| **Husband education** | | | |
| No education | 1.26 [1.19,1.35] 1.04 [0.97,1.12] | | 1.23 [1.15, 1.30] * 1.23 [0.67, 1.22] |
| Primary education | | | |
| Secondary & higher education | 1 | | 1 |
| **Maternal occupation** | | | |
| Working | 1 | | 1 |
| Not working | 0.99 [0.94, 1.04] | | 1.02 [1.01, 1.11] * |
| **Husband occupation** | | | |
| Working | 1 | | 1 |
| Not working | 0.90 [0.83, 0.97] | | 1.00 [0.92, 1.08] |
| **Religion** | | | |
| Muslim | 1 | | 1 |
| Christian | 1.62 [1.53, 1.71] | | 0.46 [0.57, 1.94] |
| Other | 1.22 [1.13, 1.31] | | 1.36 [0.75, 1.46] |
| **Contraceptive utilization** | | | |
| No | 1.11 [1.05,1.17] | | 1.03 [0.98, 1.08] |
| Yes | 1 | | 1 |
| **Place of delivery** | | | |
| Home | 1.30 [1.21, 1.40] | | 1.33 [1.24, 1.43] * |
| Institution | 1 | | **1** |
| **Antenatal Care Visit** | | | |
| No | 1.01 [0.85, 1.19] | | 1.31 [1.10, 1.55] * |
| 1–4 visit | 0.94 [0.89, 0.99] | | 1.13 [0.37, 1.19] |
| >4 visits | 1 | | 1 |
| **Media exposure** | | | |
| No | 1.14 [1.08, 1.21] | | 1.17 [1.11, 1.24] * |
| Yes | 1 | | 1 |
| **Internet utilization** | | | |
| No | 1.69 [1.58, 1.80] | | 1.68 [0.67, 1.79] |
| Yes | 1 | | 1 |

*(Continued)*

**Table 4.** (Continued)

| Individual and community level factors | Model I AOR [95% CI] | Model II AOR [95% CI] | Model III AOR [95% CI] |
|---|---|---|---|
| **Visiting health facility last 12 months** | | | |
| No | 0.92 [0.87, 0.98] | | 0.91 [0.87, 0.97] * |
| Yes | 1 | | 1 |
| **Health insurance** | | | |
| No | 1.26 [1.19, 1.34] | | 1.18 [1.11, 1.26] * |
| Yes | 1 | | 1 |
| **Wealth index** | | | |
| Poor | 2.80 [2.63, 2.97] | | 2.29 [2.14, 2.45] * |
| Middle | 1.79 [1.68, 1.92] | | 1.57 [1.46, 1.68] * |
| Rich | 1 | | 1 |
| **Community level variable** | | | |
| **Residency** | | | |
| Urban | | 1 | 1 |
| Rural | | 2.29 [2.24, 2.35] | 1.30 [1.22, 1.39] * |
| **Region SSA** | | | |
| East Africa | | 1 | **1** |
| West Africa | | 2.0 [1.92-2.04] | 2.38 [2.21, 2.55] * |
| **Community ANC visit** | | | |
| Low | | 1.00 [0.93, 1.09] | 0.93 [0.86, 1.00] |
| High | | 1 | 1 |
| **Community wealth index** | | | |
| Low | | 0.75 [0.70, 0.81] | 1.12 [1.02, 1.22] * |
| High | | 1 | 1 |
| **Community media exposure** | | | |
| Low | | 1.01 [0.92, 1.10] | 0.94 [0.86, 1.03] |
| High | | 1 | 1 |
| **Community literacy level** | | | |
| Low | | 1.10 [1.00, 1.20] | 1.13 [1.03, 1.24] * |
| High | | 1 | **1** |

**Model I:** *individual level variable,* **Model II:** *Community level variable,* **Model III**: *Individual and community level variable.*

## Discussion

Barriers to healthcare access have contributed to maternal and child mortality and morbidity in developing countries such as those in Sub-Saharan Africa. This study aimed to assess the prevalence and determinants of barriers to healthcare access among women in Sub-Saharan Africa.

In this study, 55.8 of women experienced barriers to healthcare access. This finding is lower than estimates reported in previous studies from South Africa and Tanzania 65% [18,19] and Ethiopia and sub-Saharan Africa 60–72% [1,17,20,21]. This variation could be due to changes in socioeconomic, health infrastructure, health system policies, women's attitudes towards health care seeking behavior, and cultural differences across countries in Sub-Saharan Africa. Moreover, these discrepancies might be due to differences in the study period, because this study includes the most recent DHS data sets, from 2019 to 2023. On the other hand, the findings of this study are similar to but slightly higher than in previous studies Ghana and Gambia (51%), (45.5%) [22,23]. This variation may be attributed to differences in socioeconomic conditions,

health infrastructure, and health system policies, as well as variations in women's health-seeking attitudes and cultural contexts across countries in sub-Saharan African.

The odds of barriers to healthcare access were 1.30 times higher among reproductive age women living in rural residency compared to women living in an urban area. This finding is supported by previous studies conducted in Ethiopia [1,20], Guinea [24], Gambia, Tanzania and Ghana [15,23,25], respectively and Sub-Saharan countries [22], East Africa [26]. This association may be due to the fact that rural areas have low health infrastructures, i.e., inaccessibility to health facilities, less opportunities and the influences to socio-cultural practices that oblige women to seek permission from their husbands to visit the facility because women's decisions for seeking healthcare services are under their husbands/partners control. Moreover, male involvement and support for women health care access are limited [1,14,22,26]. This implies that governments better to expand infrastructure services in a rural areas and encourage male involvement and support for improving healthcare access for reproductive age women [14].

The odds of perceived barriers to healthcare access were 1.23 and 1.21 times more likely to among reproductive age women whose husbands had no education or primary education compared to women whose husbands had secondary and above educational level, respectively. This finding was consistent with previous studies conducted in east Africa, Ethiopia, Tanzania, Benin, and Bangladesh [1,15,20,26,27]. The possible explanation could be in a better access to the healthcare among women with educated husbands might be due to better participation and involvement of husbands in their families' health [22]. Women whose partners are educated are also likely to be informed regarding their fundamental human rights and may have higher health literacy. As a result, they are more likely to deal with any form of barrier to healthcare, compared to their counterparts who are less educated and may have lower health literacy [14,15,20,28].

The odds of barriers to healthcare access were 1.33 times more likely to occur among women who delivered at home compared to women who had an institutional delivery. This evidence was supported by a study conducted in Senegal [29]. The plausible justification might be that women who delivered at home may not get health information/ counselling from healthcare providers that empower them to use other maternal healthcare services and women who deliver at home may face varous barriers to healthcare access such as financial constraint, lack of transportation, limited autonomy in healthcare decision and sociocultural norms that discourage institutional delivery. Moreover, they can be affected by the importance of enhancing health system outreach, strengthen community-based education, and addressing sociocultural and socioeconomic disparities that affect health seeking behavior. On the other hand, the odds of barriers of accessing health care among women who had no media exposure were 1.17 times more likely to compared with women who had media exposure. This evidence was supported by studies conducted in Ethiopia [30] and Nepal [31]. This could be the power of mass media in disseminating information concerning maternal health that may enhance women's knowledge and attitude towards health care access and utilization [31]. Barrier of accessing health care among woman who were not visiting health facility within 12 months were 9% times less likely compared to women who had visiting health care facility within 12 months of study period. The odds of barrier to accessing health care among woman who had no health insurance utilization were 1.18 times more likely to compared with a woman who had used health insurance. This finding was supported with study done in Ethiopia [1], sub-Saharan Africa [22], Tanzania [15], Ghana [25]. The possible reason might be due to the fact that, individuals who were not had health insurance faced for unexpected catastrophic expenditure and difficulties in obtaining money for consultation of the doctor [1]. The odds of barrier to accessing healthcare among woman who were classified under poor and middle wealth status were 2.29 and 1.57 times more likely accessing health care difficulty compared to women who were classified under rich wealth index category. This evidenced was supported with study done in Ethiopia, [1,30,32,33] Tanzania [15] Uganda [34], sub Saharan Africa [14]. This finding might be explained by the fact that high out-of-pocket expenses and a lack of accessibility to healthcare facilities pose significant obstacles for middle-class and impoverished women in sub-Saharan Africa. Many people in the region cannot afford healthcare services because over 40% of health costs are paid for directly by individuals [35–38].

The odds of barrier of accessing health care among woman who live west Africa region were 2.38 times more likely barriers to accessing health care compared to east Africa region. According a recent joint report from the World Health Organization and the United Nations states that high out-of-pocket costs, a lack of adequate healthcare infrastructure, and other sociocultural barriers make it more difficult for women of reproductive age in West African nations such as Nigeria, Mali, and Sierra Leone to access maternal healthcare. On the other hand, East African countries like Rwanda, Kenya, and Ethiopia have strengthened their community health initiatives and gender-inclusive policies, which have improved women's autonomy in accessing healthcare services [39]. Furthermore, our current findings are supported by recent Demographic and Health Survey (DHS) program data (2022–2023), which indicate that women in West African countries are 2–3 times more likely to report delays in accessing healthcare due to cost, shortages of healthcare personnel, and long distances to health facilities. In contrast, East African countries, particularly Uganda and Tanzania, have strengthened their health insurance systems and rural outreach initiatives. Additionally, the World Bank's 2023 Health Service Coverage Index reveals that women in West Africa have significantly lower access to family planning, skilled birth attendance, and other essential reproductive health services [23,40,41].

## Strengths and limitations

The main strength of this study is the use of nationally representative data, which allows for a comprehensive evaluation of contextual and individual factors associated with barriers to healthcare access. Additionally, the analysis accounted for the clustering within the sample by employing advanced statistical models, enhancing the robustness the findings.

The limitation of this study was many Sub-Saharan African countries have not conducted DHS surveys since 2019, which may affect the representativeness of recent data. In addition, health worker–related factors were not included in the analysis, potentially omitting important segments of population of healthcare access. Furthermore, future research better consider integrating objective measures, such as wearable devices, as demonstrated by Zhao et al. (2025) [42].

## Conclusions

In this study, over half of women in Sub-Saharan Africa face barriers to healthcare access, emphasizing the need for focused strategies to improve equitable service delivery. Husband educational status, media exposure, internet utilization, place of residency, ANC visit status, place of delivery, health insurance utilization status, wealth index status, visiting health facility within 12 months, community literacy level and community wealth index and country category (Western, Eastern Africa) were significantly associated with barriers of accessing health care. Therefore, the governments and ministries of health in Sub-Saharan Africa countries better taking in to consideration those factors while designing policies and strategies targeting to health care access and talking the preventable barriers to healthcare access in sub-Saharan Africa.

## Supporting information

**S1 Table. 5: Bivariable logistic regression of determinate factors associated with barriers to healthcare access among women in Sub-Saharan Africa.**
(DOCX)

**S2 Table. 6: VIF AND AIC AND BIC table.**
(DOCX)

## Acknowledgments

We would like to say the DHS programs for allowing us to use the relevant DHS dataset in this study.

## Author contributions

**Conceptualization:** Gebreeyesus Abera Zeleke, Mulugeta Wassie, Alebachew Ferede Zegeye, Tadesse Tarik Tamir.

**Data curation:** Enyew Getaneh Mekonen, Bewuketu Terefe, Alebachew Ferede Zegeye, Mohammed Seid Ali, Berhan Tekeba.

**Formal analysis:** Gebreeyesus Abera Zeleke, Enyew Getaneh Mekonen, Bewuketu Terefe, Berhan Tekeba, Tadesse Tarik Tamir.

**Investigation:** Gebreeyesus Abera Zeleke, Bewuketu Terefe, Mulugeta Wassie, Agazhe Aemro, Alebachew Ferede Zegeye, Berhan Tekeba.

**Methodology:** Gebreeyesus Abera Zeleke, Enyew Getaneh Mekonen, Mulugeta Wassie, Agazhe Aemro, Mohammed Seid Ali, Berhan Tekeba, Belayneh Shetie Workneh.

**Visualization:** Belayneh Shetie Workneh.

**Writing – original draft:** Tadesse Tarik Tamir, Belayneh Shetie Workneh.

**Writing – review & editing:** Berhan Tekeba, Belayneh Shetie Workneh.

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
