## [Decision Letter · Decision Letter 0]

27 May 2025

Dear Dr. Zeleke,

Thank you for submitting your manuscript to PLOS ONE. After careful consideration, we feel that it has merit but does not fully meet PLOS ONE’s publication criteria as it currently stands. Therefore, we invite you to submit a revised version of the manuscript that addresses the points raised during the review process.

We look forward to receiving your revised manuscript.

Kind regards,

Cristiana Abbafati, Ph.D.

Academic Editor

PLOS ONE

Journal Requirements:

Additional Editor Comments:

The present manuscript tackles a prominent and present-day public health concern—barriers to accessing healthcare services among women of reproductive age within sub-Saharan Africa—employing nationally representative DHS datasets and multilevel modeling. The study's objective is praiseworthy and has the potential to contribute to the development of policy. However, the manuscript necessitates substantial revision before publication.

Reviewers' comments:

Reviewer's Responses to Questions

**Comments to the Author**

1. Is the manuscript technically sound, and do the data support the conclusions?

Reviewer #1: Yes

Reviewer #2: Yes

2. Has the statistical analysis been performed appropriately and rigorously?

Reviewer #1: No

Reviewer #2: No

3. Have the authors made all data underlying the findings in their manuscript fully available?

Reviewer #1: No

Reviewer #2: Yes

4. Is the manuscript presented in an intelligible fashion and written in standard English?

Reviewer #1: No

Reviewer #2: No

Reviewer #1: This manuscript tackles an important public-health issue—identifying barriers to healthcare access among reproductive-age women in sub-Saharan Africa (SSA) using pooled DHS data from 2019–2023. The large sample size (N≈134,000), the inclusion of both individual- and community-level variables, and the application of multilevel mixed-effects logistic regression all strengthen its potential impact. However, before this work can be considered for publication, several critical concerns must be addressed:

1. The manuscript does not explain how DHS sampling weights, clustering, and stratification were incorporated (e.g., in Stata’s svy: framework or via robust standard errors). Please specify your approach in the Methods.

2. Multicollinearity is mentioned but actual VIF values are absent. Provide VIFs for all predictors. Include goodness-of-fit metrics (AIC, ∆Deviance) and a brief discussion of residual or influence diagnostics.

3. Several reported CIs are inverted (e.g., AOR=1.23, 95% CI: 1.30–1.15). Please correct all interval bounds and verify consistency across text, tables, and figures.

4. I recommend citing Zhao et al. (2025) “Applied statistical methods for identifying features of heart rate that are associated with nicotine vaping” . Although the outcome differs, that paper presents a robust five‐step workflow.Suggest add�“The five‐step workflow for extracting event‐related features from physiological time‐series data, originally developed by Zhao et al. ,inspired our adaptation of mixed‐effects modeling to detect ‘before/after’ patterns around reported access barriers.”

5.The manuscript contains numerous grammatical errors and non-standard expressions (e.g., “not beg problem,” “were significantly contribute”). Engage a native English speaker or professional editor to ensure clarity.

6. Tables are overly dense (Table 4 in particular). Consider splitting key results into separate tables or adding a forest‐plot figure for the main AORs.

Once these issues are adequately addressed, the manuscript will be much stronger and ready for reevaluation.

Reviewer #2: The present manuscript addresses a salient and contemporary public health concern—barriers to

accessing healthcare services among women of reproductive age in sub-Saharan Africa—utilizing

nationally representative DHS datasets and multilevel modeling. The study's objective is

praiseworthy and has the potential to contribute to the development of policy. However, in its

current state, the manuscript necessitates substantial revision (major) before it can be considered

for publication. The following section delineates the issues that should be addressed.

1. The document under review contains a multitude of grammatical inaccuracies, cumbersome

phrasing, and a lack of consistent terminology. These issues often hinder the reader's ability to

comprehend the original message. It is imperative that all documents undergo a thorough revision.

2. It is imperative that all statistical estimates undergo a thorough review to ensure internal

consistency and coherence.

3. The selection of these specific countries is not accompanied by a clear justification, nor is there

an explanation for why other countries were excluded (DHS data from 10 sub-Saharan African

countries). A rationale must be provided for the inclusion criteria, and the findings must be

explicated in terms of their generalizability to the entire SSA region.

4. The definition of the binary outcome "barrier to accessing health care" is not sufficiently

explained. The coding scheme and threshold for categorization are not clearly delineated. It is

imperative that further elucidation be provided.

5. The multilevel mixed-effects logistic regression model is a methodologicalically sound approach;

however, the following issues must be addressed: The presence of inconsistent or implausible

AORs and confidence intervals is a concern. The section comparing models is not sufficiently

clear. An analysis of the data reveals an overlap and redundancy between text, tables, and figures.

It is imperative that all model outputs are thoroughly reviewed and that any transcription errors are

rectified. The explanation of random effects statistics must be improved. It is recommended that

Figures 1 and 2 be removed or merged into a more informative graphical representation.

6. The discourse is predominantly descriptive, exhibiting a paucity of depth and analytical rigor. It is

imperative that the discussion be expanded to encompass the mechanisms underpinning the

observed associations, as well as their policy ramifications. It is imperative to differentiate distinctly

between individual, household, and system-level barriers.

7. The added value of this study is not articulated with sufficient clarity.

8. It is imperative that the keywords, table, and reference formatting be corrected.

**Do you want your identity to be public for this peer review?** For information about this choice, including consent withdrawal, please see our Privacy Policy

Reviewer #1: No

Reviewer #2: No

---

## [Author Response · Author response to Decision Letter 1]

8 Jul 2025

Reviewers and editorial team members were Comment excellent.

---

## [Decision Letter · Decision Letter 1]

28 Aug 2025

Dear Dr. Zeleke,

Thank you for submitting your manuscript to PLOS ONE. After careful consideration, we feel that it has merit but does not fully meet PLOS ONE’s publication criteria as it currently stands. Therefore, we invite you to submit a revised version of the manuscript that addresses the points raised during the review process.

Could you please carefully revise the manuscript to address all comments raised?

We look forward to receiving your revised manuscript.

Kind regards,

Helen Howard

Staff Editor

PLOS ONE

**Comments from PLOS Editorial Office** : We note that one or more reviewers has recommended that you cite specific previously published works. As always, we recommend that you please review and evaluate the requested works to determine whether they are relevant and should be cited. It is not a requirement to cite these works. We appreciate your attention to this request.

Journal Requirements:

Reviewers' comments:

Reviewer's Responses to Questions

**Comments to the Author**

Reviewer #1: (No Response)

Reviewer #3: (No Response)

2. Is the manuscript technically sound, and do the data support the conclusions?

Reviewer #1: Yes

Reviewer #3: Partly

3. Has the statistical analysis been performed appropriately and rigorously?

Reviewer #1: Yes

Reviewer #3: No

4. Have the authors made all data underlying the findings in their manuscript fully available?

Reviewer #1: Yes

Reviewer #3: Yes

5. Is the manuscript presented in an intelligible fashion and written in standard English?

Reviewer #1: Yes

Reviewer #3: Yes

Reviewer #1: The manuscript is significantly improved, and the majority of my initial concerns have been comprehensively addressed. However, one minor point remains. In the previous review, I suggested a citation that the authors acknowledged as valuable, but it has not yet been incorporated into the manuscript's body or reference list.

To potentially strengthen the discussion on limitations and future directions, the authors might consider the following point. The current analysis is comprehensive but relies entirely on self-reported data, which can be subject to recall bias. A forward-looking addition to the discussion could acknowledge this and suggest how future studies might overcome this limitation. For example, research could incorporate objective measures from technologies like wearable devices to assess certain behaviors or states. The study by Zhao et al. (2025) could be cited as an excellent example here, as it demonstrates a complete workflow for processing such complex physiological signals.

Ultimately, this is just a suggestion for the authors' consideration, and the decision to incorporate this perspective rests with them.

Reviewer #3: Major revisions are necessary, particularly with respect to the clarity of the statistical analyses and the presentation of the data.

Line 197: In the health survey data, if a household-level component exists in addition to the individual and community levels, it should usually be considered in the analysis. Are most households contribute to single observations as well as the household variance is non-negligible, and random effects are weak/unidentified? The reason(s) should be explicitly stated.

Line 196 -198: The variables included in Model I, Model II, and Model III are to be stated. Alternatively, the table footnotes should indicate the variables that correspond to each model.

Line 211- 215: The notations ‘VC’ , ‘V’, 'Variance' should be replaced with the appropriate statistical symbols. Variance null, and the variance model is to be used.

Line 214: The standard formula for MOR and not the shortcut version is to be presented.

Line 217- 218: The sentence ‘The fixed effects were employed to calculate the correlation between the probability of barriers to obtaining health care access and independent factors at the individual and community levels’ is incorrect and requires revision.

Line 223 – 225: The sentence is to be written as a method, not mentioning the results in the method section.

Line 231: The variance figures can be omitted.

Line 232: The symbol percentage for 69.8 is missing.

Table 2: The figures presented should be standardized, e.g., consistent use of commas or decimal points. The same applies to Table 3.

Line 244: Typo ‘of.3784967’. The p-value cannot be zero (to use symbol p < )

Line 263-294: It was not clear whether the data written based on Table 4, Table 5, or both. A separate write-up should be provided for each table (Table 4 referring to the association, while Table 5 refers to the determinants of barriers). It is also suggested that the manuscript be proofread thoroughly. Also, please include the crude analyses table prior to adjustment.

Table 4 & Table 5: The title not clear. Need to mention clearly or denote how the variables in Table 4 & 5 were selected.

Line 252-253: The sentence requires revision.

Line 272: In Table 4, there was no data/stem indicating higher than secondary education.

Line 273: No information on Table 5 for AOR=1.21, 95% CI: (1.31, 1.41).

Line 278: Typo 1.07 (based on Table 5).

Line 281: Typo 1.57 (based on Table 4). In Table 5, there was no data on internet utilization.

Line 293: Table 4 & Table 5: East Africa is to be used as reference.

The references do not conform to the journal’s required format.

**Do you want your identity to be public for this peer review?** For information about this choice, including consent withdrawal, please see our Privacy Policy

Reviewer #1: No

Reviewer #3: No

---

## [Author Response · Author response to Decision Letter 2]

17 Sep 2025

very important comments were given

---

## [Decision Letter · Decision Letter 2]

10 Jan 2026

Dear Dr. Zeleke,

Thank you for submitting your manuscript to PLOS ONE. After careful consideration, we feel that it has merit but does not fully meet PLOS ONE’s publication criteria as it currently stands. Therefore, we invite you to submit a revised version of the manuscript that addresses the points raised during the review process.

We look forward to receiving your revised manuscript.

Kind regards,

Alfredo Luis Fort, M.D., M.Sc., Ph.D.

Academic Editor

PLOS One

Journal Requirements:

Additional Editor Comments:

This is an important study using globally accepted data from a number of African countries on an important topic of health care utilization. However, you the authors continue to fail to produce a well-written academic English manuscript, which cannot be understood by any English-speaking reader and does not allow it to be published. I am writing to you asking to PLEASE FIND SOMEBODY WHO WRITES PROPER ACADEMIC ENGLISH before submitting your article again. I have also with extreme detail suggested lots of improvements in the attached file. Also, you cannot use the Discussion to again put each result of each barrier identified and explain it in long detail. You have to find a way to SUMMARIZE the findings so that such section is not too long. I hope you will do that in your next submission, if you want the article published. Thank you.

**Comments to the Author**

Reviewer #3: All comments have been addressed

Reviewer #4: (No Response)

2. Is the manuscript technically sound, and do the data support the conclusions?

Reviewer #3: Partly

Reviewer #4: Yes

3. Has the statistical analysis been performed appropriately and rigorously?

Reviewer #3: Yes

Reviewer #4: No

4. Have the authors made all data underlying the findings in their manuscript fully available?

Reviewer #3: Yes

Reviewer #4: Yes

5. Is the manuscript presented in an intelligible fashion and written in standard English?

Reviewer #3: Yes

Reviewer #4: No

Reviewer #3: The statement from the authors 'The household level was not included in the final model since most household contributed one individual and the variance at the household level was insignificant in the preliminary models.' is to be incorporated into the manuscript.

Reviewer #4: (No Response)

**Do you want your identity to be public for this peer review?** For information about this choice, including consent withdrawal, please see our Privacy Policy

Reviewer #3: No

Reviewer #4: **Yes:** Desalegn Areki Abay

---

## [Editor Report · Decision Letter 3]

4 Feb 2026

Barriers to healthcare access among women in sub-Saharan Africa: A pooled analysis of multi-country DHS data (2019-2023).

PONE-D-24-28999R3

Dear Dr. Zeleke,

We’re pleased to inform you that your manuscript has been judged scientifically suitable for publication and will be formally accepted for publication once it meets all outstanding technical requirements.

Kind regards,

Alfredo Luis Fort, M.D., M.Sc., Ph.D.

Academic Editor

PLOS One

Additional Editor Comments (optional):

This is the THIRD revision you have made to this manuscript. It has improved a lot and there are places where it is now well understood. Unfortunately, there are places where simply the descriptions are not well done or you have not used proper English (unfortunately, you have not found a person who can help you write but also double-check the final writing to ensure everything is well written and understood). We will see if it is possible that PLOS ONE undertakes the final "editing" of the article, because the essence, methods and results are all very valuable in this study. However, if it's not possible, then it would have to be sent to you for a "minor revision" again. If so, PLEASE FIND SOMEBODY WHO CAN FULLY REVIEW YOUR ARTICLE AND ENSURE THAT IT IS WELL WRITTEN IN PROPER ENGLISH AND IS FULLY UNDERSTANDABLE. I have made more suggestions to improve the write-up in the attached file. Thanks.

---

## [Editor Report · Acceptance letter]

PONE-D-24-28999R3

PLOS One

Dear Dr. Zeleke,

I'm pleased to inform you that your manuscript has been deemed suitable for publication in PLOS One. Congratulations! Your manuscript is now being handed over to our production team.

Kind regards,

on behalf of

Dr. Alfredo Luis Fort

Academic Editor

PLOS One